# Dual-Energy Computed Tomography-Based Iodine Concentration Estimation for Evaluating Choroidal Malignant Melanoma Response to Treatment: Optimization and Primary Validation

**DOI:** 10.3390/diagnostics12112692

**Published:** 2022-11-04

**Authors:** Hiroki Tsuchiya, Yasuhiko Tachibana, Riwa Kishimoto, Tokuhiko Omatsu, Eika Hotta, Katsuyuki Tanimoto, Masaru Wakatsuki, Takayuki Obata, Hiroshi Tsuji

**Affiliations:** 1Radiological Technology Section, Department of Medical Technology, QST Hospital, Chiba 263-8555, Japan; 2Quantum-Medicine AI Research Group, National Institutes for Quantum Science and Technology (QST), Chiba 263-8555, Japan; 3Department of Molecular Imaging and Theranostics, QST, 4-9-1 Anagawa, Chiba 263-8555, Japan; 4Department of Diagnostic Radiology and Radiation Oncology, QST Hospital, 4-9-1 Anagawa, Chiba 263-8555, Japan; 5International Particle Therapy Research Center, QST Hospital, 4-9-1 Anagawa, Chiba 263-8555, Japan

**Keywords:** choroidal malignant melanoma, contrast-enhanced CT, dual-energy CT, charged-particle radiation therapy, phantom study, clinical study

## Abstract

Contrast-enhanced imaging for choroidal malignant melanoma (CMM) is mostly limited to detecting metastatic tumors, possibly due to difficulties in fixing the eye position. We aimed to (1) validate the appropriateness of estimating iodine concentration based on dual-energy computed tomography (DECT) for CMM and optimize the calculation parameters for estimation, and (2) perform a primary clinical validation by assessing the ability of this technique to show changes in CMM after charged-particle radiation therapy. The accuracy of the optimized estimate (eIC_optimized) was compared to an estimate obtained by commercial software (eIC_commercial) by determining the difference from the ground truth. Then, eIC_optimized, tumor volume, and CT values (80 kVp, 140 kVp, and synthesized 120 kVp) were measured at pre-treatment and 3 months and 1.5–2 years after treatment. The difference from the ground truth was significantly smaller in eIC_optimized than in eIC_commercial (*p* < 0.01). Tumor volume, CT values, and eIC_optimized all decreased significantly at 1.5–2 years after treatment, but only eIC_commercial showed a significant reduction at 3 months after treatment (*p* < 0.01). eIC_optimized can quantify contrast enhancement in primary CMM lesions and has high sensitivity for detecting the response to charged-particle radiation therapy, making it potentially useful for treatment monitoring.

## 1. Introduction

Contrast-enhanced computed tomography (CT) plays an important role in the treatment of malignant tumors by facilitating the evaluation of both primary and metastatic lesions. The contrast-enhancement effect at the primary lesion increases lesion visibility, and its quantitative value is a useful index for analyzing pre-treatment pathological grade [1,2,3] and for monitoring treatment efficacy [4,5,6] in various malignant tumors. Similar merits have been suggested for the use of CT in cases of choroidal malignant melanoma (CMM) in a previous study that reported a reduction in contrast enhancement during the post-treatment course of ^125^I brachytherapy [7]. In light of this finding, contrast-enhancement analysis may be useful in the treatment planning and efficacy assessment of charged-particle radiation therapy for CMM [8,9]. This is because the decrease in tumor size is often poor, even in cases where a therapeutic benefit is realized. Therefore, the development of an index of treatment effect other than size is desirable.

However, at present, the use of contrast enhancement for CMM is mostly limited to the detection of metastatic lesions, and the technique is typically not used to assess the local response to the treatment [10]. This may be attributed to the fact that several issues need to be resolved to adequately assess CMM contrast enhancement in the clinic. First, CMM lesions generally show high density in CT and high signal intensity in T1-weighted magnetic resonance imaging (MRI) even without contrast enhancement, making it difficult to visually assess the degree of contrast enhancement. Second, quantitative evaluations based on a simple subtraction of pre- and post-contrast-enhanced images are often not adequate because of the difficulty in fixing the eye position during the exam, which can potentially cause misregistration between images. This disadvantage is especially problematic in MRI examinations, which require longer scan times. Third, for CT scanning, the pre- and post-contrast-enhancement scans double the radiation exposure to the lens in comparison with the exposure associated with a single scan after the contrast enhancement.

Dual-energy CT (DECT) may be useful to overcome these problems and adequately evaluate the contrast-enhancement effect of CMM. The three-material decomposition method [11,12] based on DECT images can estimate non-contrast-enhanced CT images and iodine concentration maps from post-contrast scans alone, utilizing the images acquired by two different energy levels in the single scan. Since the technique requires only a single CT scan, it does not involve misregistration problems. The usefulness of this approach has been well-proven in liver imaging [13,14,15], where respiratory motion was often a problem with conventional imaging approaches. In addition, the radiation exposure as well as the scan time in this approach are equivalent to those in a normal single scan, thereby avoiding an increase in the patient burden. Furthermore, the contrast-enhancement effect can be quantitatively evaluated by measuring the iodine concentration, so the problems associated with difficulty in visual assessment can be overcome. Thus, all of the above problems can be resolved.

Despite the merits of DECT, the correct estimation of iodine concentration using DECT requires a linear correlation between the CT values of the non-contrast-enhanced target tumor for the two energy levels applied in DECT (Figure 1). More specifically, when the measured CT values shown by the target tissue for the two energy levels are plotted in Figure 1, the distribution of the data needs to be linear (Figure 1, the “baseline”). Then, the estimation of the iodine concentration must be based on this specific correspondence (i.e., the formula of the “baseline”). The iodine concentration estimation is usually performed using the commercial software installed in the CT system, but unless the above prerequisite is confirmed, the estimates are inappropriate from the beginning. In addition, since the defined “baseline” in commercial software is assumed to be optimized for general human tissue rather than a specific tumor (the details are usually not specified), even if this prerequisite is fulfilled, the estimates may not be correct.

Thus, the objective of this study was to, first, experimentally confirm the above prerequisites regarding the appropriateness of estimating iodine concentration in CMM by DECT. Then, we aimed to optimize the parameters for estimation on the basis of experimental results such that accurate iodine concentration maps can be predicted for CMM. Furthermore, we aimed to evaluate the usefulness of estimating the iodine concentration in assessing the CMM response during the time course of charged-particle radiation therapy.

## 2. Materials and Methods

This retrospective observational study was approved by the local IRB of the institute that all the authors belong to, and where all the data collection and analysis were performed. The IRB waived the requirement for informed consent given the retrospective nature of the study.

### 2.1. Participants

This study consisted of two steps. The first step involved the confirmation of the prerequisite for estimating the iodine concentration and optimization of the parameters for calculation, and the second step involved the examination of the ability of the estimated iodine concentration to reflect the changes in CMM during charged-particle radiation therapy. Each step was performed using an independent cohort. The two cohorts were as follows.

Cohort 1 was a group of patients abstracted from the CMM patients treated with C^11^ charged-particle radiation therapy in 2014–2015. Those who underwent DECT examinations, including both pre- and post-contrast-enhancement scans within 1 month before the first radiation, were included in this cohort. Cohort 2 included a group of patients extracted from the CMM patients treated with C^11^ charged-particle radiation therapy in 2015–2018. This cohort included patients who had undergone post-contrast DECT scans less than 1 month before treatment, 3 months after treatment, and 1.52 years after treatment. Cases with tumor heights less than 5 mm and those showing strong calcification in the lesion were excluded from both cohorts. Finally, cohort 1 included 27 patients (M:F = 10:17, age: 16–79 years), and cohort 2 included 21 patients (M:F = 11:10, age: 27–84 years). None of the cases in cohort 2 showed local recurrence or relapse during the observation period.

### 2.2. DECT Scans

All the scans in this study were performed by the same scanner in the clinic (SOMATOM definition Flash^®^, SIEMENS, Erlangen, Germany). This device uses two independent tubes for DECT scans, with tube voltages of 80 and 140 kVp, respectively. As a result, the system outputs 80- and 140-kVp images, along with a 120-kVp image that is synthesized from the 80- and 140-kVp images (equivalent to a typical clinical CT image). In addition, the estimated iodine concentration map was obtained for each post-contrast-enhanced scan using the commercial software installed in the CT system (eIC_commercial).

Contrast enhancement was performed using a commercially available agent (Iopamidol Injection Syringes, 370 mg/mL, 100 mL; Fuji Pharma Co., Ltd., Tokyo, Japan). Patients with body weights <62 kg received a dose of 600 mg/kg over 60 s, while other patients received 100 mL of the agent at the rate of 1.6 mL/s. To suppress the eye movement as much as possible, an LED pointer was used to create the fixation point on the gantry ceiling, which the patients were instructed to stare at during the scan.

### 2.3. ROI Setting for Cohort 1

First, registration was performed for the pre- and post-contrast-enhanced CT scans of each case such that the largest cross-section of the tumor coincided. Then, a region of interest (ROI) was manually set within this matched cross-section (Figure 2a). These operations were performed using our in-house software that works on MATLAB 2017a (MathWorks, Natick, MA, USA). These ROIs were used in all the ROI-based studies for cohort 1.

### 2.4. ROI Setting for Cohort 2

A three-dimensional (3D)-ROI spanning multiple slices was manually set for the tumor area in each scan (Figure 2b). The procedure began with the selection of all the slices that included the tumor and the exclusion of the top- and the bottom-most slices to avoid the partial-volume effect. Second, for each slice, the temporary ROI was manually defined by tracing the edges of the tumor cross-section. Finally, all the temporary ROIs were combined to form the single 3D-ROI for the case. The edge of each temporary ROI was pushed as close to the tumor margins as possible but also carefully kept inside the tumor region to minimize the partial-volume effect. These 3D-ROIs were used in all the ROI-based studies for cohort 2.

### 2.5. Exam 1: Validation of the Appropriateness of Estimating the Iodine Concentration in CMM and Optimization

Cohort 1 was used for this examination. For the pre-contrast-enhancement scan in each case, the CT values of the tumor area for 80- and 140-kVp images were obtained. The CT value was defined as the average of the pixels within the ROI. Then, a linear approximation was formulated for the relationship between the 80- and 140-kVp CT values. The validity of this linear approximation was statistically tested using Pearson’s correlation test, and *p* < 0.05 was considered significant.

When the relationship between the 80- and 140-kVp CT values in the non-contrast-enhanced tumor can be approximated linearly, the line can be used as the optimized “baseline” in Figure 1 to estimate the iodine concentration for the tumor. The estimated iodine concentration based on the experimentally obtained linear correlation in the previous section was defined as the optimized iodine concentration estimate (eIC_optimized) in this study.

### 2.6. Exam 2: Validation of the Advantage of Optimized Estimates over General Estimates

Cohort 1 was used for this evaluation. To assess the accuracy of eIC_optimized in comparison with eIC_commercial, the ground truth iodine concentration (GT) was measured for each case: the average CT values before and after contrast enhancement were calculated based on ROI, and the GT was obtained by applying the difference between these two to the relationship between the iodine concentration and the CT value (obtained by an additional phantom study, not shown). Then, eIC_commercial and eIC_optimized were compared on the basis of their differences from GT. The Wilcoxon sign-ranked test was used for this comparison, and *p* < 0.05 was considered significant.

### 2.7. Exam 3: Evaluation of the Usefulness of eIC_Optimized

Cohort 2 was used for this examination. The capability to assess changes in CMM during charged-particle radiation therapy was evaluated for the following indices: tumor volume, CT values (80 kVp, 140 kVp, and 120 kVp), and eIC_optimized. The tumor volume was obtained by multiplying the number of pixels included in the 3D ROI by the pixel volume, and the values were normalized within each case such that the tumor volume before treatment was 1. The values of the indices obtained before treatment, 3 months after treatment, and 1.5–2 years after treatment were compared. The Wilcoxon signed-rank test was used for these comparisons, and *p* < 0.05 was considered significant.

## 3. Results

### 3.1. Exam 1

A graph plotting the 80- and 140-kVp CT values obtained for the non-contrast-enhanced tumor is shown in Figure 3. The data points were distributed linearly, which was also supported by the Pearson’s correlation test (R = 0.94, *p* < 0.01). The linear approximation for this correlation between 80 kVp and 140 kVp CT values was used as the “baseline” in Figure 1 to obtain eIC_optimized.

### 3.2. Exam 2

The relationships between the GT values and eIC_commercial and eIC_optimized values are shown in Figure 4a. The data points of eIC_optimized were distributed closer to the y = x line than those of eIC_commercial, indicating a more accurate estimation. This result was supported statistically, where the difference from GT was significantly smaller for eIC_optimized than for eIC_commercial (*p* < 0.01).

### 3.3. Exam 3

No recurrence or relapse was observed in all patients in cohort 2 during the observation period. Tumor volume tended to decrease after charged particle therapy, showing a significant reduction at 1.5–2 years after treatment in comparison with that before the treatment. However, the volume reduction was not significant at 3 months after treatment (Figure 5a). Similarly, CT values (80, 140, and 120 kVp) also tended to decrease after treatment, and the change was significant at 1.5–2 years post-treatment but was not significant at 3 months post-treatment (Figure 5b). On the other hand, eIC_optimized decreased significantly both between pre-treatment and 3 months post-treatment and between 3.

## 4. Discussion

In this study, we assumed that the estimation of the iodine concentration using DECT would facilitate the accurate evaluation of the contrast enhancement in CMM, and we experimentally validated the appropriateness and optimized the parameters for estimation. In addition, we assessed the clinical merit of iodine concentration estimates in comparison with measurements of tumor volume and the CT value for the detection of post-treatment changes after charged-particle radiation therapy.

First, we examined the distribution of CT values at 80 kVp and 140 kVp in non-contrast-enhanced tumors and found a strong linear correlation between the two (Figure 3). Therefore, we considered that the prerequisite for the three-material decomposition [11] is satisfied, and a reasonable iodine concentration estimation can be made.

Second, we optimized the parameters to obtain the estimated iodine concentration based on the linear correlation between the 80- and 140-kVp CT values in the CMM that was identified in the process described previously. As a result, the contrast enhancement of CMM could be estimated more accurately in eIC_optimized than in eIC_commercial, and the difference was significant (Figure 4). The major improvement of eIC_optimized over eIC_commercial may have been achieved because the calculation parameters to obtain eIC_commercial were designed to fit more general body tissue and therefore were not adequate for CMM.

In the assessment of the changes after charged-particle radiation therapy, the tumor volume and the contrast-enhanced CT value both decreased significantly over 1.5–2 years after treatment (Figure 5), indicating a favorable treatment response [16]. On the other hand, changes in tumor volume and CT values were not significant at 3 months post-treatment, whereas eIC_optimized decreased significantly even at 3 months post-treatment. These results suggest that post-treatment changes could be observed more sensitively by eIC_optimized than the other two more frequently used indices (Figure 5). The higher sensitivity of eIC_optimized was also found visually (Figure 6). This advantage may be useful for post-treatment monitoring.

Interestingly, the eIC_optimized and CT values showed different sensitivities to tumor reaction, because both eIC_optimized and CT values reflect local iodine concentration after contrast enhancement. The difference may be attributed to the fact that CT values are also influenced by substances other than iodine. For example, at 3 months post-treatment, the CT values may have been elevated by slight hemorrhage in the early post-treatment period [17] and did not correctly reflect the reduction in the contrast-enhancement effect.

The local tumor volume did not decrease at 3 months post-treatment. Since the most major factor for tumor volume reduction can be assumed to be the dropout of tumor cells after cell death [18], the tumor volume may not change immediately in the early stage after radiation treatment. In addition, edema may have occurred as a reaction to the radiation treatment [19], and the apparent volume increase due to edema may have masked the reduction in tumor volume. However, the more important point may be that even though the method used to measure the tumor volume in this study was one of the most careful of the generally possible methods, the measurement error may not have been negligible. For example, in one case, the measured volume in this study increased after 1.5–2 years (Figure 5a) but was judged to be unchanged by the physician in the clinic. The tumor in this case was relatively small before the treatment, so the measurement error may have been large. In other words, evaluations based on tumor volume are laborious but may still not be accurate enough. This aspect highlights the usefulness of eIC_optimized as an easy and quantitative index to observe the tumor reaction.

Here, even when eIC_commercial was used instead of eIC_optimized, a significant decrease was observed between the values obtained pre-treatment and 3 months post-treatment (Appendix A). Thus, within the scope of the present study, eIC_commercial can detect early post-treatment changes as well as eIC_optimized. However, since the iodine concentration estimate is a quantitative assessment, and since eIC_commercial has been shown to have imprecise estimates in comparison with eIC_optimized (Figure 4), eIC_commercial should not be considered sufficient.

This study had several limitations. First, since the number of cases was small, eIC_optimized may have room for further improvement. Further optimization and validation in independent cohorts with a sufficient sample size are needed in this regard. Second, pathological evaluation was not performed in this study since we focused on post-treatment changes. Therefore, the reason for the greater sensitivity of eIC_optimized over CT values is not evident. Considering the absence of recurrence or relapse in all observed patients, further studies are required to assess the capability of eIC_optimized as a tool for the CMM treatment, including whether eIC_optimized is overestimating the post-treatment changes or not. Third, the clinical merit of monitoring eIC_optimized in CMM patients was not definitively proven because all of the patients had a favorable outcome. Further studies involving patients with both favorable and unfavorable outcomes are needed.

## 5. Conclusions

The present study first validated the appropriateness of estimating iodine concentrations for CMM by using DECT and then optimized the parameters to calculate the estimate. The results showed that the estimated iodine concentration was more sensitive for the detection of post-charged-particle radiation therapy changes in CMM than the standard methods, suggesting that this approach may be useful for monitoring the local response during treatment.

## Figures and Tables

**Figure 1 diagnostics-12-02692-f001:**
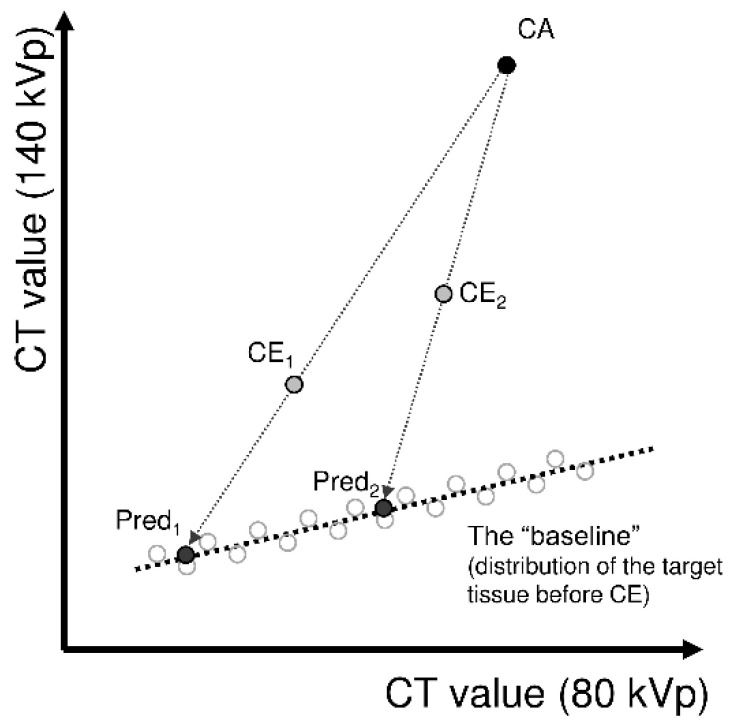
Overview of iodine concentration estimation from post-contrast-enhanced dual-energy computed tomography (DECT). When the CT values for the two energy levels of DECT can be assumed to be linearly correlated in the target tissue without contrast enhancement (the “baseline”), the measured CT values after contrast enhancement (CE_1_, CE_2_) can be considered to have deviated from somewhere on the baseline toward the point of iodine contrast agent (CA). Therefore, the CT values before contrast enhancement can be predicted (pred_1_, pred_2_) by obtaining the intersection of the baseline and the line connecting CA and CE_1_, CE_2_, respectively. In addition, iodine concentration can be estimated by determining the difference between CE_1_ and pred_1_, as well as that between CE_2_ and pred_2_.

**Figure 2 diagnostics-12-02692-f002:**
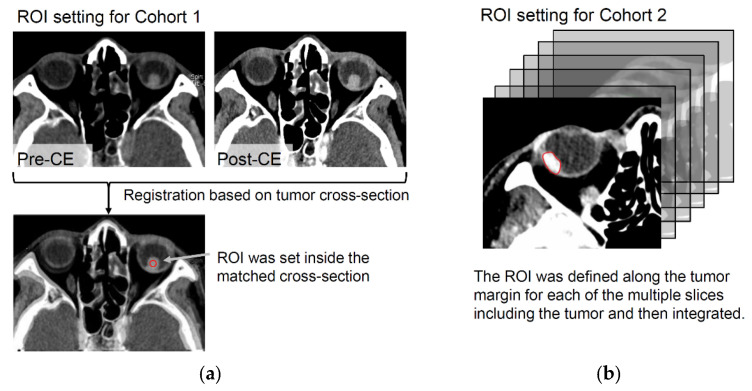
Region of Figure 2. Region of interest (ROI) definition for cohorts 1 and 2. (**a**) In cohort 1, to define a similar ROI for the pre- and post-contrast-enhanced images, the slice with the largest cross-section of the tumor was extracted from the pre- and post-contrast-enhanced series, and the slices were registered according to the tumor cross-sections. Then, the ROI was defined inside the matched tumor cross-section. (**b**) In cohort 2, ROIs were defined along the tumor margins for each slice including the tumor (the top and bottom slices were excluded), and they were integrated to form a single three-dimensional ROI.

**Figure 3 diagnostics-12-02692-f003:**
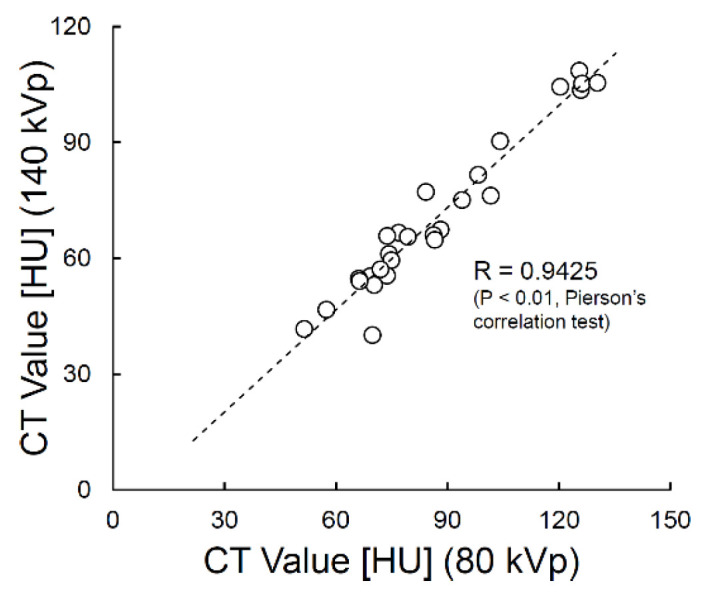
Distribution of computed tomography (CT) values in choroidal malignant melanoma before contrast enhancement. A strong linear correlation was found between the 80- and 140-kVp CT values. The iodine concentration can be estimated adequately from post-contrast-enhanced dual-energy CT using this correlation.

**Figure 4 diagnostics-12-02692-f004:**
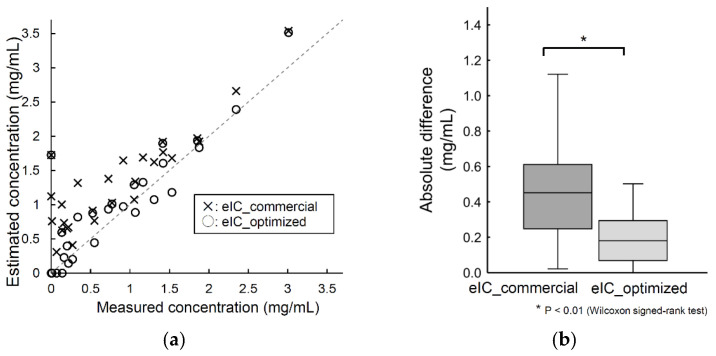
Improved accuracy through optimization of iodine concentration estimation. (**a**) The relationship between the ground truth iodine concentration (GT) obtained from the difference between pre- and post-contrast images, the estimated iodine concentrations obtained by the software equipped in the computed tomography system (eIC_commercial), and the optimized iodine concentration measurement developed in this study (eIC_optimized). eIC_optimized shows a distribution closer to that of GT than eIC_commercial. (**b**) The difference from GT was significantly smaller in eIC_optimized than in eIC_commercial.

**Figure 5 diagnostics-12-02692-f005:**
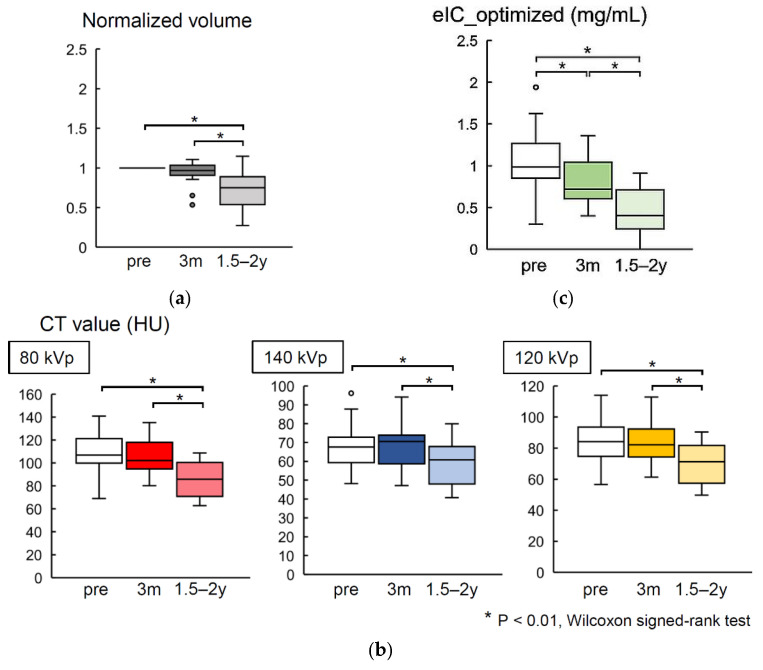
The changes in indices during charged-particle radiation therapy. In comparison with the pre-treatment values, the (**a**) tumor volume and (**b**) computed tomography values (80, 140, 120 kVp) significantly reduced at 1.5–2 years post-treatment, but none of these parameters showed significant changes at 3 months after treatment. (**c**) On the other hand, the eIC_optimized showed significant reductions at both 3 months post-treatment and 1.5–2 years post-treatment.

**Figure 6 diagnostics-12-02692-f006:**
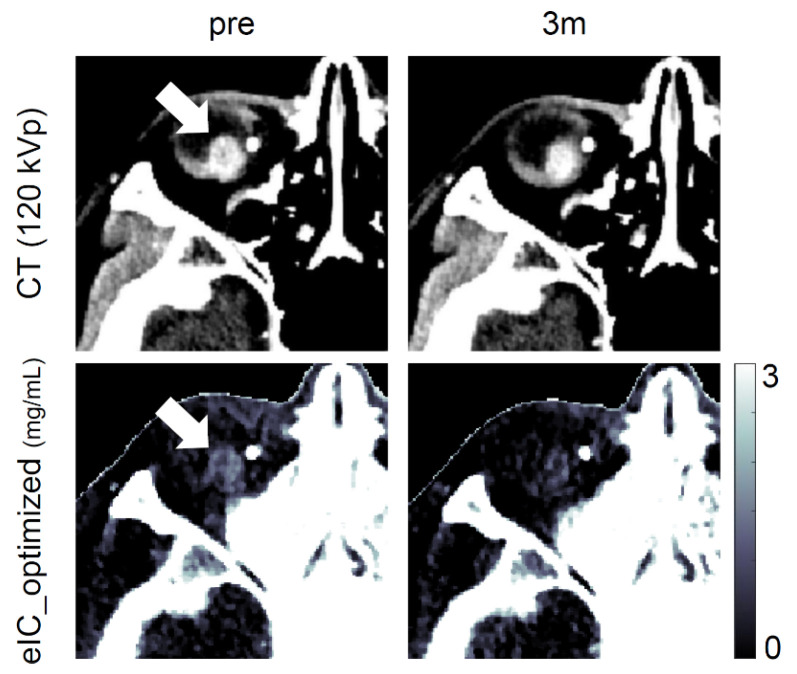
The contrast-enhanced CT 120 kVp and eIC_optimized map images in the representative case (pre-treatment and 3 months after treatment). The arrows indicate the tumor. Despite the favorable outcome that was judged years after, the change in the tumor cannot be judged by CT images at 3 months after treatment (the density of the tumor seems even elevating at 3 months in this specific case). On the other hand, the months and 1.5–2 years post-treatment (Figure 5c). The images of the CT 120 kVp and eIC_optimized map before and after 3 months after treatment in a representative case are shown in Figure 6. The eIC_optimized images suggest a decrease in iodine concentration during this period, but the change is not seen in CT images. eIC_optimized map shows an obvious decrease in the estimated iodine concentration at 3 months compared to pre-treatment. The bony areas and the metallic marker (white dot at the left of the tumor) are shown in white color in the eIC_optimized images. The iodine concentration in these areas, which must be zero or very small, cannot be estimated correctly because eIC_optimized is specialized for calculating iodine concentration in a tumor.

## Data Availability

The data presented in this study are available on request from the corresponding author. The data are not publicly available due to the ethical concerns for patient privacy.

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
