# Peer review of "Dual-Energy Computed Tomography-Based Iodine Concentration Estimation for Evaluating Choroidal Malignant Melanoma Response to Treatment: Optimization and Primary Validation"

_diagnostics, 2022, doi:10.3390/diagnostics12112692_

Round 1

Reviewer 1 Report

Authors described the optimization and primary validation of DECT-based iodine concentration estimation regarding choroidal malignant melanoma (CMM).   Authors are comparing eIC_commercial (commercially available software) and eIC_optimized method which authors developed for floor-up of treated CMM.   THe following issues would be improved.   1) Regarding Fig. 1 (perhaps), some part of legend is missing. 2)  Fig 5(c) indicated only eIC_optimized showed reduction of tumor at 3 month after treatment. In addition, the authors described this reduction is true and other methods are not enough to detect size-reduction of tumor. However, is this true? Is there any possibility that eIC_optimized showed over-estimation.  3) Regarding above-2), can authors represent any representative cace who showed the eIC\optimized is really indicate the tumor-reduction.

Reviewer 2 Report

The manuscript is written well and the studies were well designed. The topic is relevant to the field. In the Introduction, the Authors included all the necessary information and presented the necessity of the research properly. The Authors have adequately explained the purpose of conducting the studies. The manuscript clearly described the procedure of the performed studies. The results were presented clearly and were discussed in an appropriate manner. The conclusions are supported by the results presented in the manuscript.
The references and the figures are appropriate. However, the description of Figure 2 needs formatting improvements, because the first words are missing.
I recommend the manuscript for publication after making edit according to the following comment. 

Round 2

Reviewer 1 Report

Authors modified their manuscript according to the reviewers' comments.